# Advancements in the Impact of Insect Gut Microbiota on Host Feeding Behaviors

**DOI:** 10.3390/genes15101320

**Published:** 2024-10-14

**Authors:** Yikang Wang, Liang Wang, Di Li, Zhenfu Chen, Yang Luo, Juan Zhou, Bo Luo, Rong Yan, Hui Liu, Lingjun Wang

**Affiliations:** 1Department of Parasitology, Zunyi Medical University, Zunyi 563000, China; wykang1011@163.com (Y.W.); m17586533144@163.com (L.W.); li18856700782@163.com (D.L.); m19985468519@163.com (Z.C.); yang18212755200@163.com (Y.L.); zj412175@163.com (J.Z.); luobozmu@163.com (B.L.); yanrong@zmu.edu.cn (R.Y.); 2NHC Key Laboratory of Parasite and Vector Biology, National Institute of Parasitic Diseases, Chinese Center for Diseases Control and Prevention, Shanghai 200025, China

**Keywords:** gut microbiota, gut bacteria, host insects, feeding behaviour, olfactory system

## Abstract

With the application and development of high-throughput sequencing technology, the structure and function of insect gut microbiota have been analysed, which lays a foundation for further exploring the intricate relationships between gut microbiota and host feeding behaviour. The microbial community in the insect gut, as an important ecological factor, affects the host’s food selection and nutritional metabolic processes through various mechanisms, which play a key role in population dynamics and ecosystems. The implications of these interactions are profound, affecting agricultural practices, biodiversity, and the broader environment, such as pollination and pest control. In-depth exploration of the molecular mechanism of the interaction between gut microbiota and hosts contributes to the grasp of insect biology and evolution and offers novel avenues for manipulating insect behaviour for practical applications in agriculture and environmental management. This paper focuses on the possible mechanisms of insect gut microbiota regulating host feeding behaviour. It inspires further research on the interaction between gut microbiota and insects affecting host behaviour.

## 1. Introduction

Insects, as the species with the most successful survival strategies on the earth, are related to their ecological diversity and niche differentiation, as well as their strong fecundity, small size, high energy utilisation efficiency, and rapid evolutionary rate [1]. The ecological diversity and niche differentiation of insects have largely shaped the composition and function of their gut microbiota, which in turn further enhances the adaptability of insects to specific niches and plays an indispensable role in the ecological network [2]. As the main gathering place of insect gut bacteria, the midgut undertakes most of the insect’s nutrient absorption and decomposition and has a rich enzyme system [3]. Studies have shown that in addition to affecting the nutritional metabolism, the midgut microbiota is also involved in the host’s physiological, ecological, and evolutionary processes of the host, forming a close symbiotic relationship with the host [4,5].

In addition, recent studies have revealed that bacteria in the insect gut can also significantly influence host behaviour. Exploring the complex interactions between microbes and their hosts broadens our understanding of animal–microbe symbiosis and provides new perspectives and potential application strategies in agricultural pest management, biological control, and human health [6]. Therefore, this paper provides a systematic review of insect gut microbiota in terms of its identification and main functions. It also focuses on the possible mechanisms by which insect gut microbiota modulate host feeding behaviour, thus providing insights for further exploration of the specific elements of gut microbiota that influence host behaviour.

## 2. Advances in Identification Techniques of Insect Gut Microbiota

Identifying insect gut microbiota is a multi-step process, including sample collection, microbial isolation and culture, and identification by molecular biology techniques or mass spectrometry (Figure 1). Commonly used culture methods for gut microbiota include bacterial culture, high-pressure anaerobic culture microcapsule double antibody sandwich, and other culture techniques. The morphological characteristics of the colonies are observed and recorded after the culture to carry out preliminary identification [7]. However, due to the high complexity and diversity of most insect gut microbiota, modern research has increasingly favoured non-culture molecular biology techniques for analysing and identifying microbial diversity. For example, polymerase chain reaction (PCR), which amplifies the 16S rRNA gene using universal primers and then sequences the amplified products to identify microbial species [8]; 16S rDNA fingerprinting technique for the analysis of bacterial community structure based on the patterns of bands produced by electrophoresis after PCR amplification [9,10]; gene chip technology, which designs probe targeting a specific microbial species or taxon to identify the microbial composition of a sample by hybridisation; and gene microarray technology, which designs probe targeting a specific microbial species or taxon to identify the microbial composition of a sample by hybridisation. Gene microarray technology to determine the microbial composition of a sample by hybridisation, and macro-genome sequencing technology to sequence the total DNA of all microorganisms in a sample in a high throughput manner to gain a comprehensive understanding of the structure and functional genes of microbial communities without the need for pre-cultivation, etc. [11,12]. In addition, other novel techniques, such as single-cell Raman imaging, in situ hybridisation, etc. [13,14].

According to the influence of gut microbiota on the host, it is divided into probiotics, neutral bacteria, and pathogenic bacteria. In addition, gut microbiota can be further categorized according to population size and source. For example, according to the number of culturable bacteria, they are divided into major (dominant) flora and secondary flora. The major (dominant) refers to bacteria with large numbers or high population densities in the gut microbiota, typically above 10^7^~10^8^ CFU/g, and includes specialised anaerobic bacteria such as *Bacteroides*, *Eubacterium*, *Bifidobacterium*, *Ruminococcus,* and *Clostridium*, which are usually derived from the original flora [15]. Secondary flora with counts below 10^7^~10^8^ CFU/g, mainly aerobic bacteria or facultative anaerobic bacteria, such as Escherichia coli and Streptococcus, are highly mobile and potentially pathogenic, and most belong to the flora of origin or passing flora [13].

## 3. Differences in Gut Microbiota between Insects Reared in the Laboratory for Multiple Generations and Wild Insects

Insects reared in captivity in the laboratory are not exposed to bacteria in their natural habitat, including microorganisms that benefit the insect’s life activities. The antimicrobial agents in the diet of captive insect larvae and the pH reduce the likelihood of larval contamination with harmful microorganisms. Still, this situation largely reduces the chance of horizontal transmission of beneficial organisms. In other studies, it has also been found that wild *Triatoma infestans* have a greater diversity of hindgut bacteria compared to laboratory-cultured insects, the microbial composition of *T. infestans* in the laboratory does not reflect the complete collection of gut microbes in wild *T. infestans*, and extrapolating the gut microbiota profiles of wild insects by studying laboratory insects alone may not be possible [16]. The general assumption is that microbial diversity contributes to the vital activities of insects; however, insects raised artificially in laboratory environments are, perhaps, less functionally diverse due to a reduction in bacterial diversity and increased susceptibility [17].

## 4. Main Functions of Gut Microbiota and Possible Mechanisms Influencing Host Feeding Behaviour

The gut microbiota of insects plays multiple important functions in host insects [18,19,20]. There are complex and close interactions between insect gut microbiota and their hosts, involving regulation of nutrient metabolism, growth and reproduction, and feeding behaviour regulation [21,22] (Figure 2), and in-depth studies of these functions and the possible mechanisms of their feeding behaviours are of great significance to the understanding of insect ecology and microbial interrelationships.

### 4.1. The Role of Gut Microbiota in Digesting Food

Insect gut microbiota play multiple roles in their digestion of food (Figure 3). In phytophagous insects, the microbial community in the gut can degrade cellulose, e.g., microbes in the gut of the termite help the host break down lignocelllose [23] efficiently. Some insects specialise in pollen as their main source of nutrition. The pollen grains are protected by a shell that the insect must overcome to gain access to abundant nutrients. Bees work in close collaboration with their intestinal commensal bacteria, which encode a series of enzyme genes, such as pectin-degrading enzymes, glycoside hydrolases, and polysaccharide hydrolases, which help bees efficiently utilise the carbohydrate resources of pollen [24,25,26].

In addition, gut microbiota exhibit potential food detoxification functions, as the effective nutritional value of many food sources relies on their non-toxic state, and certain plant cell wall components undergo microbial-mediated hydrolysis to both detoxify and convert them into nutrients that can be fully utilised by the host [27,28]. In particular, the gut microbiota of insects that feed on toxic plants play a crucial role in processing and detoxifying such food toxicity, ensuring that insects obtain the energy and nutrients they need to survive from a unique and complex food chain [29]. Many insects avoid plant tannins during feeding, which reduces the protein content of the food. Many microorganisms produce tannase, an enzyme that allows insects to be more adapted to local food sources [30].

### 4.2. Gut Microbiota Influences Nutritional and Metabolic Functions of Insects

Gut microbiota plays an indispensable role in insect digestion and nutrient absorption processes [31,32,33]. Intestinal microorganisms also play an important role in insect hosts, such as participating in insect growth and development, adapting to the environment, and drug resistance [34]. They not only help the host insect to decompose and transform food components efficiently but also provide a variety of essential nutrients such as vitamins, amino acids, and nitrogen sources and are deeply involved in the anabolic process of these substances [35,36,37].

Moreover, the gut microbiota of insects can enhance the recycling of urea and uric acid, aid insects in fending off infections from foreign pathogenic microorganisms to sustain internal balance and support the host’s normal physiological functions. Insects are also rich in microorganisms on their surfaces, manure, and growth environments, which can affect their physiological functions [38]. The relationship between the growth environment of insects and the host gut microbiota needs more research and exploration.

### 4.3. Influence on the Growth, Development, and Reproduction of Host Insects

Gut microbiota largely influence the growth and development process of the host through direct interaction with host cells and regulation of host endocrine hormone signalling pathways. Experimental results have shown that *Drosophila* larvae cultured in a sterile environment exhibit significant differences, such as slowed growth rates and developmental delays, compared to *Drosophila* larvae with a normal microbiome [39,40]. However, when these sterile *Drosophila* larvae were re-implanted with appropriate amounts of *Acetobacter pomorum*, their developmental rates and growth levels could return to control standards. In addition to *Drosophila*, the gut microbiota of other non-model organisms has also been found to function in ways that affect the biological characteristics of host insects. In the case of *Drosophila Suzuki*, the complete replacement of brewer’s yeast adversely affected female fecundity, pupal weight, adult survival, and recovery rates, thus suggesting that *Enterobacter* sp. AA26 cannot fulfil the protein requirements of *D. suzukii* when yeast is absent from the diet. Interestingly, the partial yeast replacement did not present severe effects (apart from the adult recovery rate), suggesting that halving the yeast quantity is still sufficient to produce fit adults. Results in *Ceratitis capitata* indicated that the addition of *Enterobacter* sp. AA26 increased pupae and adult production and decreased rearing duration for several developmental stages. *Enterobacter* sp. AA26 proved to be an adequate nutritional source for *C. capitata* larvae [41]. The *Enterobacter* sp. AA26 gut symbiont has been shown to shorten immature developmental stages, increase fecundity, extend survival, and improve male mate competitiveness and female mate acceptance in *Mediterranean fruit fly* [42]. Alterations in gut microbiota dynamics and their impact on the feeding and growth of *Spodoptera frugiperda* following antibiotic treatment are being investigated. Antibiotic treatment suppresses the gut bacteria of *Spodoptera frugiperda*, leading to decreased food intake and body weight of the larvae and an extension in the developmental period [43]. Insect gut microbiota exerts diverse influences on insect reproduction, including enhancing reproductive rates, boosting reproductive efficiency, altering reproductive modes, and modifying the adaptability of offspring [44]. Additional studies have similarly revealed the importance of gut microbiota for aphid growth and reproduction: aphids showed a decrease in growth, developmental rate, and reproductive capacity when antibiotics were used to eliminate microbes from the aphid gut [45]. Similar effects have been demonstrated in stink bugs (*Megacopta punctatissima*), where the elimination of gut microbiota by antibiotics resulted in stagnant development, sterility, and a significant increase in mortality of stink bug individuals [46]. Additionally, alterations in the gut microbiota can stimulate the host’s immune mechanisms to resist pathogen invasion and influence insect development [47,48]. This further confirms that gut microbiota is critical for healthy growth and survival at all insect life cycle stages [49].

### 4.4. Regulating the Adaptation of Insects to Their Environment and Host Plants

The conditions of their surroundings largely govern the life activities of insects, and among the many environmental factors, the temperature has a particularly significant effect [50]. The range of temperatures that insects can tolerate and adapt to, and their resilience to change, is a central element in determining their geographic distribution patterns. A large body of evidence reveals the important role of symbiotic microorganisms on the environmental adaptability of insects [51], among which the effect of endosymbiotic bacteria on the heat tolerance of insects is a central focus of research in this field, especially the interrelationships between endosymbiotic bacteria and their insect hosts, which has become a central research element in understanding how insects respond to global climate change, especially the increase in temperature [52]. These endosymbiotic bacteria can be passed from one generation to the next within the insect and directly influence host physiology, such as metabolism, immune responses, and tolerance thresholds to temperature extremes [53]. It has been shown that the diversity of secondary bacteria in aphids increases when aphids feed on a wider variety of host plants. It is inferred that symbiotic flora may expand the range of host plant lineages that host insects can adapt to and utilise by influencing their ability to utilise different plant resources.

### 4.5. Influence of Gut Bacteria on Host Insect Feeding Behaviour and Possible Mechanisms

The influence of gut bacteria on the feeding behaviour of host insects has become a hotspot at the intersection of ecology, microbiology, and neurobiology in recent years [54]. The mechanisms of their influence may involve the production and regulation of host neurotransmitters, hormone levels, nutrient intake, and the insects’ olfactory systems.

#### 4.5.1. Influence on Host Insect Neurotransmitter Synthesis and Secretion

Gut microbiota plays a key role in the regulation of neurotransmitters in insects. Neurotransmitters, as key chemicals for inter-neuronal signalling [55], play an important role in the regulation of appetite in insects. Studies have shown that gut microbiota can modulate insect feeding behaviour by influencing neurotransmitter synthesis and host release mechanisms [56]. For example, certain gut microbiota may promote the production and release of specific neurotransmitters, thereby enhancing the insect’s appetite; conversely, other microorganisms may inhibit these processes, decreasing the insect’s desire to feed. Jia Y et al. [57] showed that food intake could be controlled by regulating the expression level of octopamine in *Drosophila*. When the activity of this neurotransmitter was inhibited, *Drosophila* feeding was significantly reduced. In addition, octopamine, as a neurotransmitter widely found in the arthropod nervous system, is involved in and regulates various physiological functions and is particularly important in connecting the physiological state of insects with their feeding behaviours. A series of experiments by Selcho M et al. [45] further confirmed the effect of octopamine on the feeding behaviours of insects. They found that octopamine positively stimulated feeding activity and that this stimulatory effect was closely related to the concentration of octopamine in the insect brain. Octopamine regulates foraging behaviour and food intake by activating specific neuronal pathways related to feeding behaviour.

#### 4.5.2. Hormone Levels Affecting Host Insects

The gut microbiota community interacts with the endocrine system of insects through diverse mechanisms, which in turn affects their feeding behaviour [46]. Specifically, gut bacteria can modulate energy use and the desire to feed by regulating insect hormone levels, such as glucagon, insulin and other hormones related to energy metabolism and appetite [58]. 20-Hydroxyecdysone is a moulting hormone commonly found in insects, and it has a variety of bioactive functions. This hormone interacts with dopamine receptors and significantly affects feeding behaviour and pupation. Kang et al. [59] revealed that 20-Hydroxyecdysone binds to dopamine receptors in Lepidoptera, inhibiting feeding and facilitating entry into the pupal stage. They further explored the mechanism behind this interaction, i.e., the binding of 20-Hydroxyecdysone to the dopamine receptor activates specific signalling pathways, modulating neuronal activity, ultimately leading to reduced feeding and enhanced pupation behaviour. DopEcR (a G-protein-coupled dopamine/ecdysterone receptor), as a dual G-protein-coupled receptor, recognises the catecholamine hormone dopamine and also responds to the steroid hormone ecdysteroid. In *Drosophila*, ecdysone and dopamine are involved in the stress response [60]. In the agricultural pest *Helicoverpa armigera*, DopEcR is a key switch integrating dopamine and 20-Hydroxyecdysone (20E) signalling, which determines whether larvae will continue to remain actively feeding and growing or switch to sedentary behaviour and begin to enter the pupal stage [38]. 20-Hydroxyecdysone concentrations are low in the feeding stage of larvae and high in the preparation for the pupal stage. In contrast, dopamine concentrations were higher during the active feeding period of larvae and decreased when they were about to pupate.

#### 4.5.3. Influencing Nutrient Intake of Host Insects

Gut microbiota has a significant impact on the selectivity of host nutrient absorption, and the insect gut microbiota synergistically regulates food digestion in the host through various mechanisms [61,62]. These microorganisms secrete key digestive enzymes that break down complex macromolecules in food into smaller molecular fragments that are more readily absorbed, thus significantly affecting the ability of insects to acquire and utilise desired nutrients from food, which in turn may indirectly influence their food selection preferences [30]. Li et al. [63] found a significant effect of *Exorista japonica* on the intestinal microbial composition of *Bombyx mori*. After the parasitic wasps infected the silkworms, the gut microbiota structure of the silkworms changed significantly, resulting in a decrease in the efficiency of feed digestion and absorption, a slowdown in growth, and a decrease in feed conversion efficiency. This may be because the parasitic wasps inhibited the growth of beneficial flora in the silkworm’s intestine by some means while promoting the proliferation of potentially harmful flora, which disrupted the original intestinal microbial equilibrium. Experiments comparing the differences between bacterial *Drosophila* and normal *Drosophila* carrying gut microbiota when ingesting the same food revealed that gut bacteria facilitate the absorption of specific nutrients such as proteins and fats in *Drosophila* [64]. The important findings of this series of studies indicate that the impacts of gut bacteria on host nutrient absorption are closely related to their regulation of the dynamic balance of the gut microbiota community, further emphasising the centrality of gut microbiota–host interactions in the nutrient absorption process and their potential applications.

#### 4.5.4. Effects on the Host’s Olfactory System

Several studies have revealed that insect gut microbiota may influence the host olfactory system by modulating the expression of host olfactory receptor genes [56,65,66]. These microbes can influence host foraging behaviour directly or indirectly through modulation of brain function by mediating the supply of essential amino acids or by producing a variety of compounds that interfere with insect feeding preferences [67]. Gut microbiota not only guide food choice by modulating amino acid metabolism but also produce a variety of chemicals that intervene in insect feeding behaviour. Gut-derived compounds such as short-chain fatty acids, bile acids, methylamines, amino acid derivatives and microbe-associated molecular patterns stimulate energy metabolism and regulate food intake through signalling mechanisms [68]. The gut microbiota shape host foraging behaviour to some extent through metabolic activities, and these metabolic pathways alter the effectiveness of nutrients (and their derivatives) recognised by the central nervous system. In addition, bacterial tryptophan metabolism is one of the key pathways by which gut microbiota influence host behaviour [69], as tryptophan, a major regulatory molecule for the synthesis of the central neurotransmitter 5-hydroxytryptamine (5-HT), has been shown to drive foraging behaviour and dietary choices in several experimental studies [70].

Some microorganisms can also produce chemical signals that insects can perceive, thus influencing their selection and localisation of host plants. Volatile organic compounds (VOCs), as a chemical signal transmitted through the air, have a significant effect on the olfactory perception and feeding behaviour of insects [71,72]. Studies have shown that volatile organic compounds produced by gut microbiota can alter the ability to perceive food odours, affecting insect food preferences [56,73]. For example, specific VOCs may direct insects to prefer foods rich in particular nutrients. Gut bacteria of the genus Citrobacter in *Drosophila orientacea* release 3-hexenyl acetate (3-HA), which attracts other female fruit flies that choose to lay their eggs on the same host fruit [74]. On the other hand, the gut bacteria of the bark beetle *Dendroctonus valens* produce a multifunctional pheromone called verbenone that helps the beetle determine whether pine trees are suitable for settlement [75]. The volatile organic compounds produced by gut bacteria are used for mutual attraction between conspecifics and to build reciprocal relationships between different organisms. Plants release specific plant-induced volatiles (HIPVs) when fed on by herbivorous insects. This process has a direct effect on herbivorous insects and also provides indirect defence by attracting natural enemies [76]. Insect gut bacteria also influence this indirect defence mechanism. For example, VOCs released from leaves act as attractants for natural enemies in the indirect defence response triggered by the rice brown planthopper (*Nilaparvata lugens; BPH*), while the symbiotic bacterium *Hamiltonella defensa* has been shown to reduce the recruitment rate of parasitic wasps by reducing the number of volatile compounds produced by the plant, thereby increasing the recruitment rate of pea aphid (*Acyrthosiphon pisum*) and survival adaptations. Despite extensive research on the impact of gut microbiota on host insect behaviour, the studies have primarily focused on a limited range of insect species. Additionally, laboratory conditions fail to fully replicate the complex interactions present in natural environments, which may render the applicability of the findings to real-world scenarios somewhat limited [77].

## 5. Conclusions

Significant breakthroughs have been made in studying the influence of gut microbiota on the feeding behaviour of insect hosts. Current scientific evidence suggests that microorganisms in insects play a crucial role in shaping their food preferences and feeding strategies by optimising their nutrient metabolism, interfering with neurophysiological signalling mechanisms, and regulating immune responses and overall physiological status. With the development and application of techniques for the study of gut microbiota, more research has been conducted on gut microbiota. However, more in-depth studies are needed to investigate how gut microbiota can adjust and transform the feeding behaviour of the host. Future research pathways could focus on the following directions: firstly, to clarify the causal links between specific gut bacterial species and their metabolites and host-specific eating habits and to identify key strains or metabolites with significant regulatory effects. Second, the molecular mechanisms behind microbe–host interactions should be systematically analysed using integrated multi-omics analyses (including transcriptomics, proteomics, and metabolomics). In addition, it is necessary to explore how environmental factors (e.g., food quality, temperature changes, pollutant exposure, etc.) affect the gut microbiota and change the feeding behaviour of the host and to assess the importance of such dynamic interactions between micro-ecology and macro-ecology in terms of the ecological adaptive capacity of the species and the geographical distribution pattern. In the long run, these research results are expected to be applied to integrated pest management, effective use of biological resources, and environmental protection. By precisely regulating insect gut microbiota, we can improve insect population management strategies and promote the development of sustainable agricultural practices while safeguarding ecosystem health and stability. Some gut bacteria help pests adapt to host plants and degrade phytotoxins. Thus, the symbiotic relationships between gut bacteria and insect pests have partly contributed to the success and diversification of insects but have also increased the difficulty of pest control. Interestingly, some studies have found that insect gut bacteria can potentially be manipulated to increase or decrease insect fitness [73]. Gut bacteria have great potential as biocontrol agents for pest management. In addition, biological control agents cause less pollution to the environment compared to chemical insecticides. However, considering that most enteric bacteria are not culturable and other species may be present in insect–plant interactions in the field, laboratory studies may have limitations, making future research difficult.

## Figures and Tables

**Figure 1 genes-15-01320-f001:**
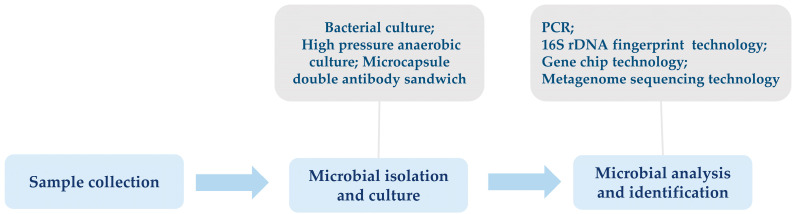
The basic process of insect gut microbiota identification.

**Figure 2 genes-15-01320-f002:**
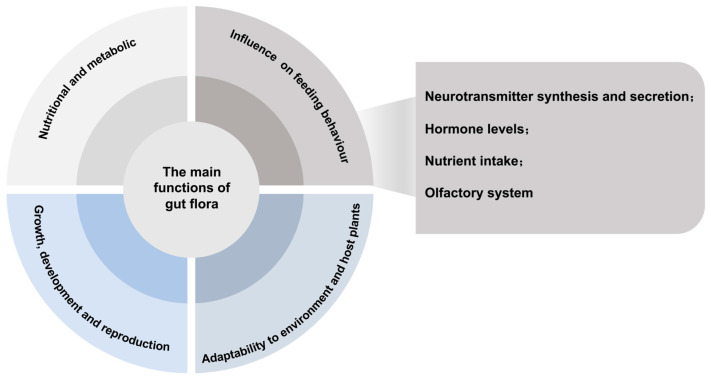
The main functions of gut microbiota and the possible mechanisms affecting host feeding behaviour.

**Figure 3 genes-15-01320-f003:**
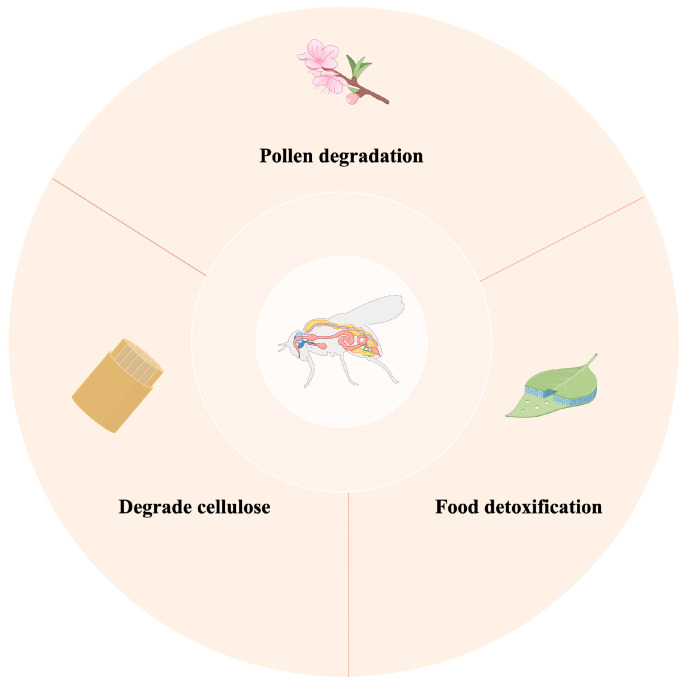
The role of gut microbiota in digesting food.

## Data Availability

No new data were created or analyzed in this study.

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
