# Peer review of "Advancements in the Impact of Insect Gut Microbiota on Host Feeding Behaviors"

_genes, 2024, doi:10.3390/genes15101320_

Round 1
Reviewer 1 Report
Comments and Suggestions for Authors
The paper entitled “Advancements in the impact of insect gut microbiota on host feeding behaviors” addresses the importance of gut microbiota on insect growth, development, and reproduction. Gut microbiota has been proven to paly a role on insect physiology and they can have crucial implications in the pest management approaches. The manuscript is well-written and covers a broad range of gut microbiota role in insect ecology and physiology. I have only some minor comments which need to be addressed before publication.
1. The manuscript is a review paper that presents collectively data of other studies without discussing their limitations and their experimental design. Authors can elaborate more on this aspect.
2. It is also important to include more studies on non-model organisms apart form Drosophila. Insects that are of economic importance and they are targeted from pest management programs should be included (e.g. https://pubmed.ncbi.nlm.nih.gov/31870292/ , https://pubmed.ncbi.nlm.nih.gov/34680692/ )
3. The conclusions part is very short. Authors need to elaborate more and discuss there the importance of gut microbiota. For example in lines 300-301 there is only a brief and general reference on integrated pest management that needs to be discussed more with additional references supporting this argument.
4. All species names should be in italics.
5. There are several grammatic errors in the text. Please correct them or ask a native English speaker to do it.
6. The format of the text contains a lot of errors. There are no gaps where there should be (e.g. the reference number is almost always attached to the last word), and vice versa.
7. In lines 152-160 references are required.
8. L239: Affecting Effects on the host's olfactory system
Comments on the Quality of English Language
Minor editing of English language required.
Reviewer 2 Report
Comments and Suggestions for Authors
The manuscript written by Yikang Wang, Liang Wang, Di Li, Zhenfu Chen, Yang Luo, Juan Zhou, Bo Luo, Rong Yan and Hui Liu, Lingjun Wang, „Advancements in the impact of insect gut microbiota on host feeding behaviors” is a review that presents the current knowledge in the field regarding the insect gut microbiota.
The text is clear and easy to read.
The literature used is adequate.
I encourage its publication after minor editing.
Reviewer 3 Report
Comments and Suggestions for Authors
I believe that the paper "Advancements in the impact of insect gut microbiota on host feeding behaviors" is well-written and informative. However, it would greatly benefit from a section that discusses the differences in microbiota between insects that have been reared in a laboratory for many generations and their wild counterparts. This is important because it directly reflects how the difference in their microbiota not only affects feeding behavior but also significantly impacts the fitness of the insects. I recommend that the authors reorganize the paper to include a section on "microbiota activity on food," making it clear that microorganisms play a crucial role in breaking down cellulose and its derivatives in food, as well as aiding insects in detoxification. Although the authors mention these points, they seem to get lost in the text. To address this, the paper could be divided into more sections, and a figure could be included to visually depict the functions and benefits of the microbiota in the feeding of insects.
